

# Development and validation of a personalized classifier to predict the prognosis and response to immunotherapy in glioma based on glycolysis and the tumor microenvironment

Pengfei Fan[1],[*], Jinjin Xia[1],[*], Meifeng Zhou[1], Chao Zhuo[2] and Hui He[2]

[1] Department of Neurology, Changxing People's Hospital, Huzhou, Zhejiang, China
[2] Department of Pediatrics, Changxing People's Hospital, Huzhou, Zhejiang, China
[*] These authors contributed equally to this work.

## ABSTRACT

**Background:** Glycolysis is closely associated with cancer progression and treatment outcomes. However, the role of glycolysis in the immune microenvironment, prognosis, and immunotherapy of glioma remains unclear.

**Methods:** This study investigated the role of glycolysis on prognosis and its relationship with the tumor microenvironment (TME). Subsequently, we developed and validated the glycolysis-related gene signature (GRS)-TME classifier using multiple independent cohorts. Furthermore, we also examined the prognostic value, somatic alterations, molecular characteristics, and potential benefits of immunotherapy based on GRS-TME classifier. Lastly, the effect of kinesin family member 20A (KIF20A) on the proliferation and migration of glioma cells was evaluated *in vitro*.

**Results:** Glycolysis was identified as a significant prognostic risk factor in glioma, and closely associated with an immunosuppressive microenvironment characterized by altered distribution of immune cells. Furthermore, a personalized GRS-TME classifier was developed and validated by combining the glycolysis (18 genes) and TME (seven immune cells) scores. Patients in the GRS$^{low}$/TME$^{high}$ subgroup exhibited a more favorable prognosis compared to other subgroups. Distinct genomic alterations and signaling pathways were observed among different subgroups, which are closely associated with cell cycle, epithelial—mesenchymal transition, p53 signaling pathway, and interferon-alpha response. Additionally, we found that patients in the GRS$^{low}$/TME$^{high}$ subgroup exhibit a higher response rate to immunotherapy, and the GRS-TME classifier can serve as a novel biomarker for predicting immunotherapy outcomes. Finally, high expression of KIF20A is associated with an unfavorable prognosis in glioma, and its knockdown can inhibit the proliferation and migration of glioma cells.

**Conclusions:** Our study developed a GRS-TME classifier for predicting the prognosis and potential benefits of immunotherapy in glioma patients. Additionally, we identified KIF20A as a prognostic and therapeutic biomarker for glioma.

Corresponding author
Hui He, cxhehui@163.com

## INTRODUCTION

Glioma is a prevalent primary malignant tumor within the intracranial region, accounting for approximately 80% of all malignant brain tumors (*Sampson et al., 2020*). Glioma is classified into four grades by the World Health Organization (WHO): Grades I–IV. Grades I and II are classified as low-grade glioma (LGG), whereas Grades III and IV are classified as high-grade glioma (HGG) (*Berger et al., 2022*). Grade IV glioma is specifically known as glioblastoma (GBM). The median survival time of GBM is approximately 14–15 months, and the 5-year survival rate is below 5% (*Ma, Taphoorn & Plaha, 2021*). Glioma treatment primarily involves surgical resection, complemented by radiotherapy, chemotherapy, and other comprehensive treatment approaches (*Bush, Chang & Berger, 2017*; *Nicholson & Fine, 2021*). However, due to the substantial heterogeneity and malignant progression of glioma, recurrent and progressive manifestations of glioma are common during treatment failure. Even after treatment, LGG tend to recur and progress to HGG or even evolve into glioblastoma. Molecular biomarkers play a crucial role in personalized treatment and the assessment of clinical prognosis for glioma patients (*Sledzinska et al., 2021*). Previous studies have identified specific molecular biomarkers, including isocitrate dehydrogenase (IDH) mutation, co-deletion of chromosome arms 1p and 19q (1p/19q codeletion), $O^6$-methylguanine-DNA methyltransferase (MGMT) promoter methylation, and telomerase reverse transcriptase (TERT) promoter mutation (*Yan et al., 2009*). *Eckel-Passow et al. (2015)* reported that gliomas were classified into five principal groups based on *1p/19q*, *IDH*, and *TERT* promoter mutations. In addition, several gene signatures have been established to predict the prognosis of glioma (*Zhang et al., 2020*, *2022a*, *2022b*). It is well known that the prognosis of glioma is influenced by various factors, including histologic type, grade, and molecular subtypes. However, owing to the heterogeneity of glioma, the predictive potential of these biomarkers may vary among individual patients. Therefore, there is a critical need for the identification of novel prognostic and therapeutic biomarkers in patients with glioma.

The tumor microenvironment (TME) of glioma exhibits significant heterogeneity and plays a crucial role in glioma development, occurrence, and the efficacy of immunotherapy (*Gieryng et al., 2017*; *Pombo Antunes et al., 2020*). Tumor cells secrete a variety of chemokines that promote the infiltration of different immune cells, including macrophages, bone marrow-derived suppressor cells, CD4[+] T cells, and regulatory T cells (Tregs) (*Guo & Wang, 2023*). These interactions between cytokines, chemokines, and extracellular matrix components reprogram infiltrating immune cells, resulting in an immunosuppressive microenvironment that drives glioma progression. Immunotherapy approaches that target molecules within the immune microenvironment, including immune checkpoint blockade and immune cell therapy, have been developed and demonstrated effectiveness in treating various types of cancer (*Adachi & Tamada, 2015*; *Mougel, Terme & Tanchot, 2019*; *Waldman, Fritz & Lenardo, 2020*). However, the

effectiveness of CTLA-4 and PD-1 inhibitors in glioma is still limited due to the complexity of the immunosuppressive environment (*Genoud et al., 2018*; *Liu et al., 2020*). Glioma faces multiple obstacles to immunotherapy, including immune cell dysfunction and tumor-associated immune inhibitory factors. Additionally, the presence of the blood-brain barrier presents challenges for drug penetration into tumor tissue (*Steeg, 2021*). Therefore, investigating the molecular mechanisms underlying glioma development and the immune microenvironment not only enhances our understanding of glioma pathogenesis but also improves the sensitivity of glioma to immunotherapy.

Currently, several biomarkers have been identified to guide immunotherapy response prediction, including PD-L1 expression, tumor-infiltrating lymphocytes, tumor mutation burden (TMB), and mismatch repair deficiency. However, their clinical utility is still limited by inter-tumor heterogeneity, dynamic changes in expression, and the complex interplay within the tumor microenvironment. This study systematically analyzes the cancer hallmarks and the immune microenvironment of glioma. Our research findings highlight that glycolysis serves as a crucial prognostic risk factor in glioma and is associated with an immunosuppressive microenvironment. Furthermore, we developed a glycolysis-related gene signature (GRS)-TME classifier that utilizes glycolysis-related genes and the TME-related immune cells, thereby improving risk stratification and prediction accuracy for patients with glioma. We have studied correlations between the GRS-TME classifier and gene mutations, molecular signaling pathways, as well as immune markers. The GRS-TME classifier serve as guidance for prognosis management and decision-making regarding immunotherapy for patients with glioma. Finally, we have identified and experimentally validated a glycolysis-related key gene, kinesin family member 20A (KIF20A), which serves as a prognostic biomarker and therapeutic target for glioma.

## MATERIALS AND METHODS

### Collection and processing of transcriptomic data

For this study, we obtained four glioma-related transcriptomic cohorts from The Cancer Genome Atlas (TCGA) and the Chinese Glioma Genome Atlas (CGGA) database, including TCGA-LGG, TCGA-GBM, CGGA-325, and CGGA-693. A total of 1,661 glioma patients were included for further analysis after excluding those with incomplete survival information. The RNA sequencing data underwent normalization by converting the raw counts to transcripts per million (TPM) and applying log2 (TPM+1) transformation. Furthermore, the TCGA-LGG and TCGA-GBM cohorts were combined to create the Meta-TCGA cohort, which was used to develop a prognostic GRS-TME classifier for glioma patients. Similarly, the CGGA-325 and CGGA-693 cohorts were combined to form the Meta-CGGA cohort, which was used to validate the prognostic model. To eliminate batch effects between the datasets, we applied the ComBat algorithm from the *sva* package.

### Analysis of cancer hallmarks and tumor immune microenvironment

To examine the effect of cancer hallmarks on the prognosis in glioma, we collected 29 cancer hallmarks gene sets from the Molecular Signatures Database (MSigDB) (*Liberzon*

*et al., 2015*). These gene sets were then utilized to analyze the effects on glioma prognosis through single-sample gene set enrichment analysis (ssGSEA) and univariate Cox regression analysis. We employed the ESTIMATE algorithm to compute the stromal score, immune score, and tumor purity of glioma patients (*Yoshihara et al., 2013*). Furthermore, the CIBERSORT algorithm was utilized to estimate the composition of immune cells in the tumor immune microenvironment of glioma patients (*Newman et al., 2015*). Additionally, Tumor Immune Dysfunction and Exclusion (TIDE) is a computational tool used to assess immune dysfunction and exclusion (*Jiang et al., 2018*). By employing TIDE, we computed the TIDE score of glioma patients and predicted their potential response to immunotherapy. We also used the Imvigor210 cohort to assess the effect of the GRS-TME classifier on patients' response to immunotherapy.

## Gene mutation and functional enrichment analysis

Glioma gene mutation data was extracted from the TCGA database. We performed subsequent analysis using the *maftools* package to examine the mutational spectrum and TMB. To elucidate potential mechanisms linked to the GRS-TME classifier, we conducted an analysis of Kyoto Encyclopedia of Genes and Genomes (KEGG) pathways and cancer hallmarks by using the *clusterProfiler* and *enrichplot* packages in R software. Results were considered significantly enriched if they had a false discovery rate (FDR) < 0.05.

## Construction of GRS score, TME score, and GRS-TME classifier

We firstly performed univariate Cox regression analysis to select genes associated with glycolysis. Using the least absolute shrinkage and selection operator (LASSO) Cox regression analysis, we developed a GRS score for glioma patients using the glycolysis-related genes. The GRS score was calculated using the following formula: GRS Score = $\sum_i$ Coefficient (mRNAi) $\times$ Expression (mRNAi). Furthermore, we identified immune cells associated with the tumor microenvironment (TME) using the CIBERSORT results. Similarly, the TME score was calculated based on the TME-related cells selected through LASSO analysis. Subsequently, patients were stratified into two groups based on the median GRS score and TME score, respectively. We subsequently developed a novel GRS-TME classifier by integrating the GRS score and TME score. Patients were further categorized into the following subgroups based on the GRS-TME classifier: GRS[low]/TME[high], Mixed (GRS[low]/TME[low] and GRS[high]/TME[high]), and GRS[high]/TME[low]. The predictive accuracy of the GRS-TME classifier was assessed and validated in different glioma cohorts and clinical subgroups.

## Cell culture and plasmids transfection

The glioma U251 cell line was obtained from the BeNa Culture Collection (Shanghai, China). Cells were cultured in Dulbecco's Modified Eagle's Medium (DMEM) containing 10% fetal bovine serum (FBS), 100 U/mL penicillin, and 0.1 mg/mL streptomycin at 37 °C in 5% $CO_2$. Transient transfections of small interfering RNA (siRNA) were performed using Lipofectamine 3000 (Thermo Fisher Scientific, Waltham, MA, USA) according to the manufacturer's instructions. The efficiency of transient transfection was evaluated

using Western blot analysis. siRNAs were chemically synthesized from GenePharma Co. (Shanghai, China) with the following sequences: Control-siRNA: 5′-UUCUCCGAACGUGUCACGUTT-3′, KIF20A-siRNA: 5′-GTTCTCAGCCATTGCTAGC-3′.

## Western blot analysis

Total protein was extracted from U251 cells and quantified using the BCA protein concentration kit (Thermo Fisher Scientific, Waltham, MA, USA). Subsequently, equal amounts of proteins were separated by electrophoresis on SDS-PAGE gels and transferred onto nitrocellulose membranes. Immunoblotting was performed using the following antibodies: Rabbit anti-KIF20A (ab70791, 1:2,000 dilution) and anti-GAPDH (ab8245, 1:500 dilution) at 4 °C overnight. Next, the membranes were washed repeatedly and then incubated with secondary antibody. The Western blot bands were visualized and analyzed by ImageJ following standard methods.

## Cell proliferation and migration assays

Cell proliferation was assessed using a Cell Counting Kit-8 (CCK-8) kit (Dojindo, Beijing, China) with approximately $2 \times 10^3$ cells seeded in each well of a 96-well plate. Following a 72-h incubation period, CCK-8 reagent was added and incubated for 1 h. The sample was measured at 0, 24, 48, and 72 h. For cell migration assays, the cell groups were resuspended in medium, and after 24 h of incubation, the transwell chambers (Corning, Corning, NY, USA) were removed. The invading cells were then fixed with 4% paraformaldehyde and stained with 0.1% crystalline violet. Finally, the transwell chambers were inverted, and the cells were photographed under a microscope.

## Statistical analysis

Statistical analysis was conducted using R software (v4.2.2). Survival analysis was performed using Kaplan-Meier curves and the log-rank test provided by the R packages *survminer* and *survival*. Student's t-test was used for statistical comparisons between two groups, and one-way analysis of variance (ANOVA) was used for statistical comparisons involving three or more groups. Multivariate analysis was performed using the Cox multivariate regression model with a stepwise method. Statistical significance was defined as $P < 0.05$ (two-tailed test).

# RESULTS

## Glycolysis as a major prognostic risk factor in glioma

To investigate the primary prognostic risk factors in glioma, we analyzed the impact of 29 cancer hallmarks on survival time using the ssGSEA algorithm and meta-analysis. The results of four glioma cohorts revealed that 22 cancer hallmarks were significantly correlated with prognosis. Among them, glycolysis exhibited the most substantial impact on the survival of glioma patients (Fig. 1A, Table S1). Notably, significant differences in glycolysis ssGSEA scores were observed among different WHO grades, demonstrating a substantial increase in glycolysis ssGSEA score among high-grade patients compared to

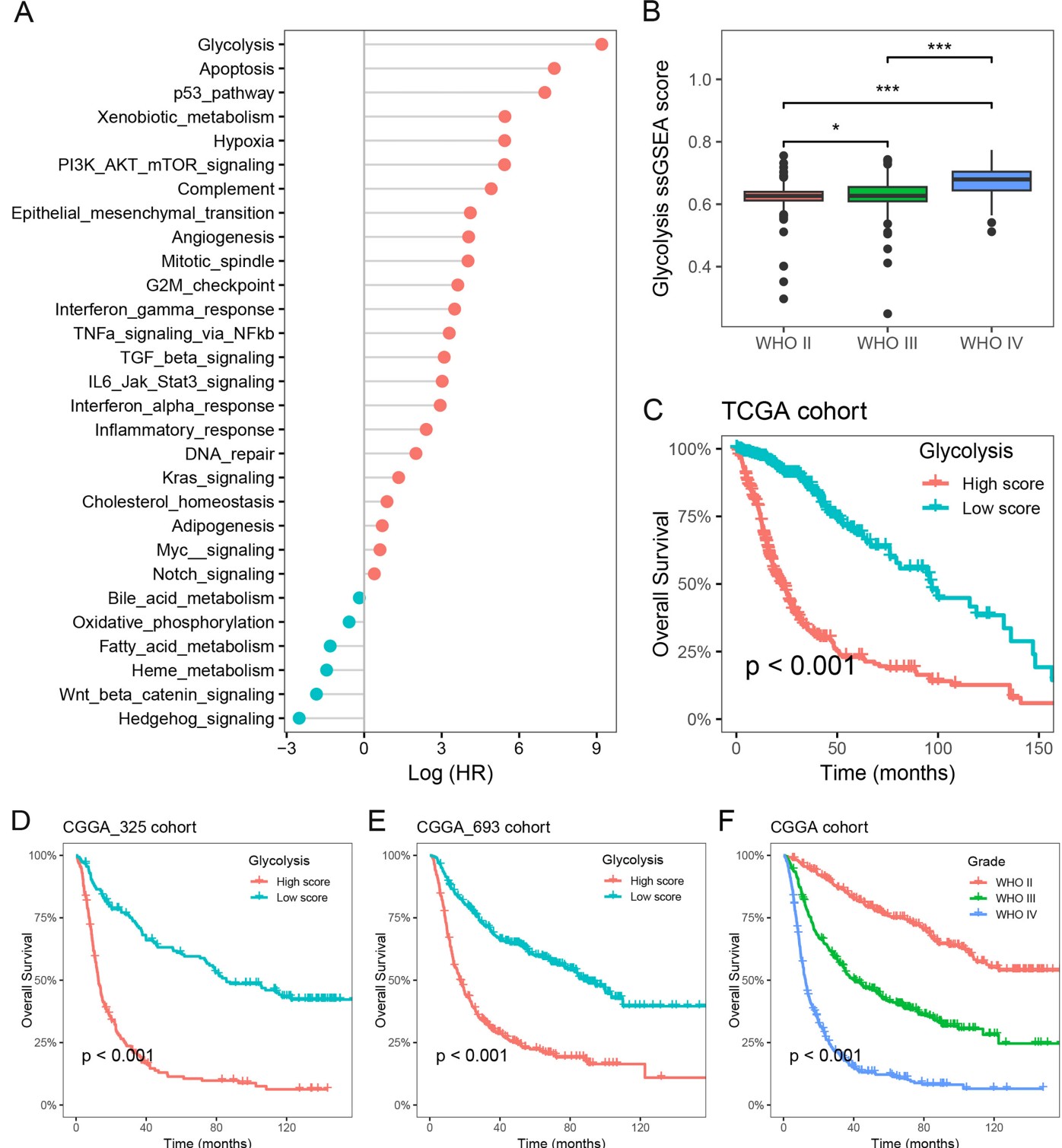

**Figure 1 Glycolysis was identified as a major prognostic risk factor in glioma.** (A) The effects of cancer hallmarks on survival prognosis in glioma; (B) differences in glycolysis ssGSEA score among pathological grade in glioma; (C) survival analysis of glycolysis ssGSEA score in the Meta-TCGA cohort; (D and E) survival analysis of glycolysis ssGSEA score in the CGGA-325 and CGGA-693 cohorts, respectively; (F) survival analysis of pathological grades in the Meta-CGGA cohort. $*p < 0.05$; $***p < 0.001$.

those with low-grade glioma (Fig. 1B). Subsequently, the glioma patients in the Meta-TCGA cohort were stratified into low- and high-groups based on glycolysis ssGSEA score. Survival analysis demonstrated that patients with low glycolysis ssGSEA score exhibited more favorable prognosis in contrast to those with high glycolysis ssGSEA score ($p < 0.001$, Fig. 1C). These findings were further validated in the CGGA_325 and CGGA_693 cohorts, consistently demonstrating that patients with high glycolysis ssGSEA score had poorer prognosis in comparison to patients with low glycolysis score (Figs. 1D and 1E). Moreover, notable differences in prognosis were observed based on WHO grades, with grade IV glioma patients exhibiting the most unfavorable prognosis (Fig. 1F). These findings strongly indicate that glycolysis as a prognostic risk factor in glioma.

## Glycolysis promotes the immunosuppressive microenvironment in glioma

We firstly examined the association between glycolysis and tumor stromal score, immune score, and tumor purity. The findings revealed that patients with high glycolysis score exhibited comparatively elevated tumor stromal and immune scores in contrast to those with low glycolysis score (Figs. 2A and 2B). However, patients with high glycolysis score had relatively lower tumor purity, indicating a higher tumor heterogeneity in patients with high glycolysis score (Fig. 2C). Furthermore, an analysis of three prominent immunosuppressive cells (CAFs, MDSCs, TAMs) revealed relatively heightened levels of CAFs and TAMs in the high glycolysis group compared to the low glycolysis group (Fig. 2D). To further explore the effect of glycolysis on the immune microenvironment, we employed the CIBERSORT algorithm to analyze the distribution of 22 immune cell types in glioma. The results demonstrated that M2 macrophages were the dominant immune cell population, exhibiting higher levels in the high glycolysis score group in comparison to the low glycolysis score group. Conversely, B cells, monocytes, activated natural killer (NK) cells, and helper T cells exhibited relatively lower levels in the high glycolysis score group (Fig. 2E). These results indicate that glycolysis modulates the immune microenvironment, and promotes the development of an immunosuppressive microenvironment in glioma.

## Development of GRS-TME classifier for improved prognostic assessment

We next assessed the prognostic significance of 189 glycolysis-related genes and 22 immune cells in glioma. Our findings indicated that 41 glycolysis-related genes and seven immune cells was significant prognostic factors ($p < 0.05$; Table S2). Utilizing these prognostic genes and immune cells, we developed GRS and TME score through the implementation of the LASSO Cox regression analysis algorithm. The detailed GRS-related genes and TME-related cells are listed in Table S3. Patients were categorized into two subgroups based on the median values of the GRS and TME scores, respectively. Notably, patients with lower GRS score displayed a more favorable prognosis in contrast to those with higher GRS score (Fig. 3A). Likewise, patients in the high TME score group demonstrated prolonged survival time (Fig. 3B). Based on the results above, we integrated the GRS score and the TME score to establish the GRS-TME classifier, which classifies

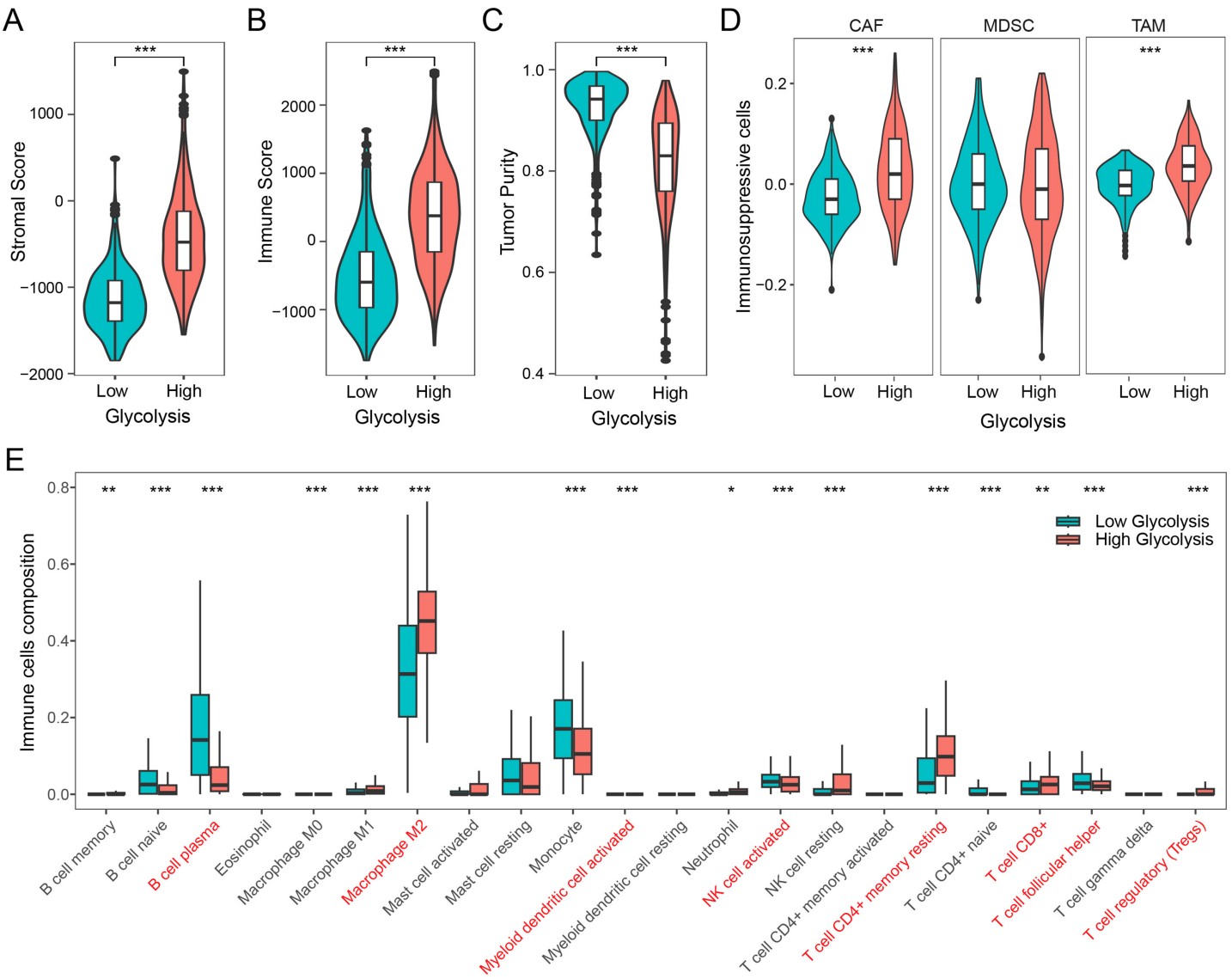

**Figure 2 Glycolysis promotes the immunosuppressive microenvironment in glioma.** (A–C) Differences in stromal score, immune score, and tumor purity among different glycolysis subgroups; (D) differences in immunosuppressive cells CAF, MDSC, and TAM among different glycolysis subgroups; (E) analysis of the content of 22 immune cells among different glycolysis subgroups. $^*p < 0.05$; $^{**}p < 0.01$; $^{***}p < 0.001$.

glioma patients into four distinct subgroups: GRS$^{low}$/TME$^{high}$, GRS$^{low}$/TME$^{low}$, GRS$^{high}$/TME$^{high}$, and GRS$^{high}$/TME$^{low}$. The GRS-TME classifier exhibited statistically distinct prognostic outcomes within the Meta-TCGA cohort, with the GRS$^{low}$/TME$^{high}$ subgroup manifesting the most favorable prognosis in comparison to the other three subgroups (Fig. 3C). Since the prognosis of patients in the four subgroups were less divergent, we merged the GRS$^{low}$/TME$^{low}$ and GRS$^{high}$/TME$^{high}$ subgroups into a mixed subgroup (Fig. 3D). By utilizing the GRS-TME classifier, glioma patients were clearly classified into three subgroups (Fig. 3E). Moreover, the GRS-TME classifier demonstrated area under the curve (AUC) values of 0.871, 0.912, and 0.864 for 1-, 3- and 5-year overall survival,

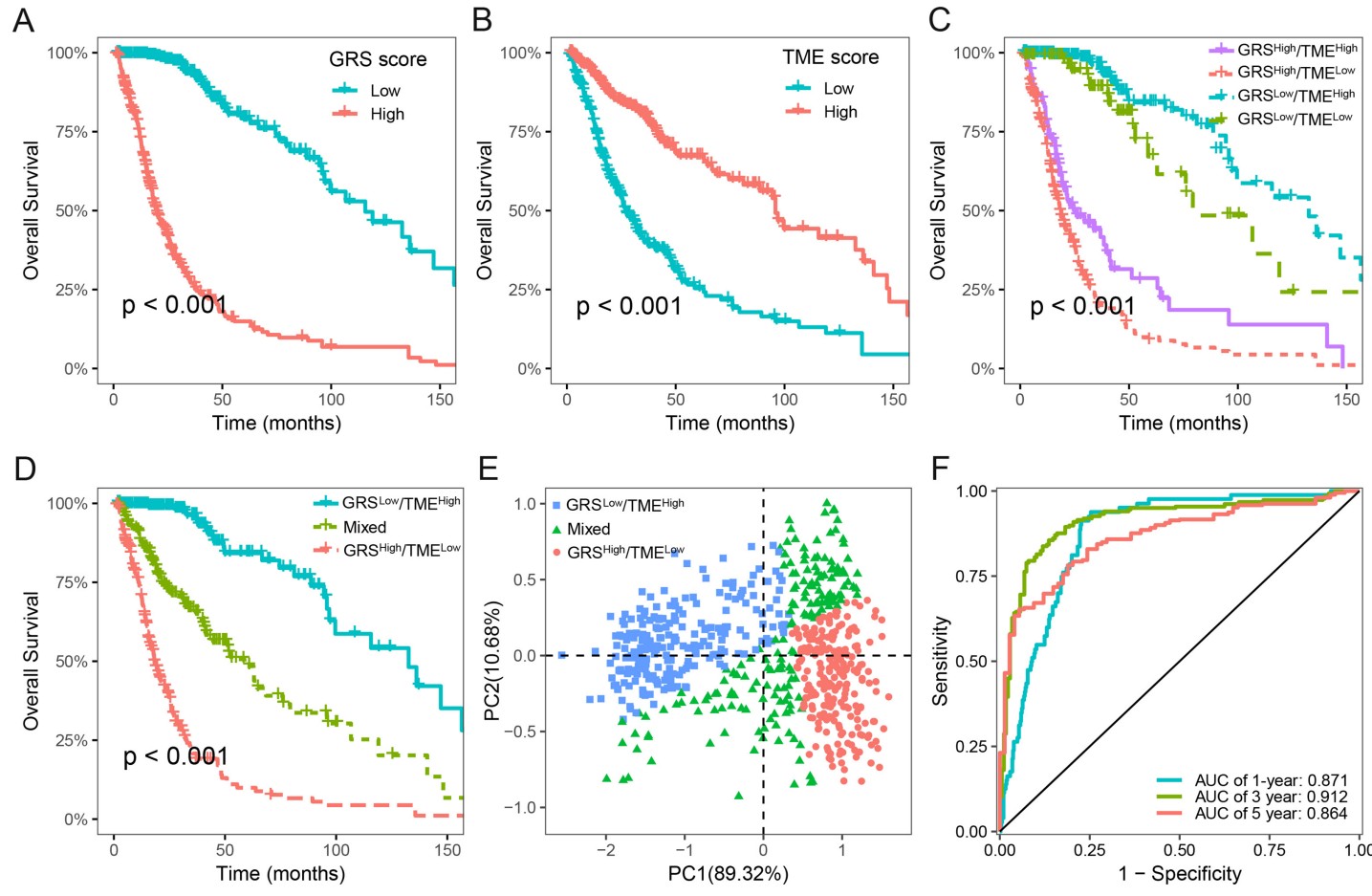

**Figure 3 Development of GRS-TME classifier for improved prognostic assessment in glioma.** (A) Kaplan–Meier survival analysis of GRS score; (B) Kaplan–Meier survival analysis of TME score; (C) Kaplan–Meier survival analysis of four different subgroups based on GRS-TME classifier; (D) Kaplan–Meier survival analysis of three different subgroups based on GRS-TME classifier; (E) two-dimensional clustering analysis based on GRS-TME classifier; (F) ROC analysis for the 1-, 3-, 5-year survival according to GRS-TME classifier.

respectively (Fig. 3F). These findings indicate that the GRS-TME classifier has the potential to enhance the accuracy of prognostic prediction in glioma.

## Validation and evaluation of the GRS-TME classifier in different cohorts

The prognostic value of the GRS-TME classifier was further validated and evaluated in the CGGA cohort. Patients in the GRS$^{low}$/TME$^{high}$ subgroup exhibited the most favorable prognosis compared to the other subgroups, which consistent with previous findings ($p < 0.001$; Fig. 4A). ROC analysis revealed AUC values of 0.705, 0.765, and 0.777, respectively, for the performance of the GRS-TME classifier at 1, 3, and 5 years, respectively (Fig. 4B). Moreover, comparable survival outcomes were observed in the CGGA-325 and CGGA-693 cohorts ($p < 0.001$; Figs. 4C and 4D). Further subgroup analysis revealed significant prognostic differences in the GRS-TME classifier among IDH wild-type and mutant type, particularly demonstrating significant prognostic stratification among patients with IDH mutant ($p < 0.001$; Figs. 4E and 4F). Pathological WHO grade

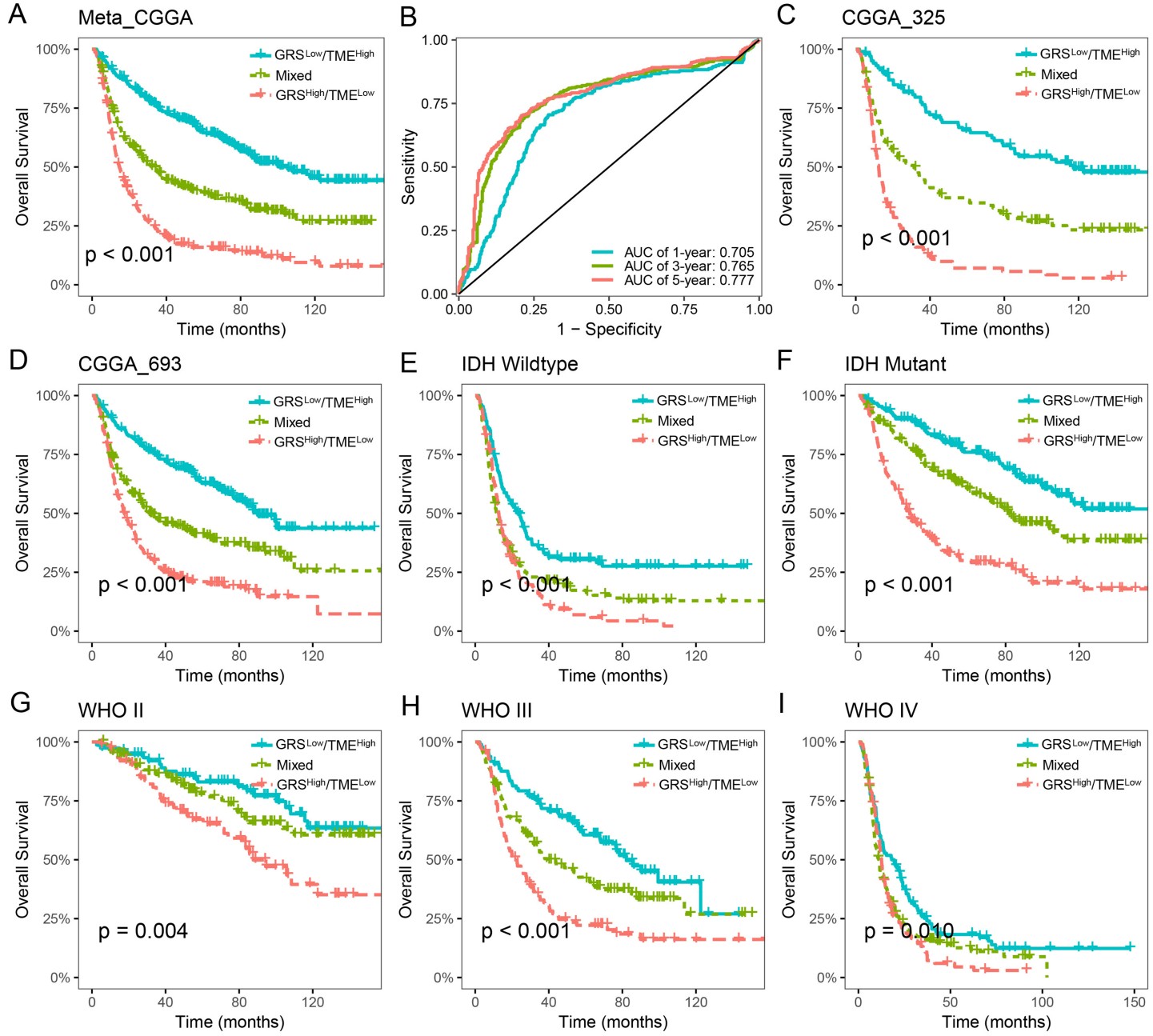

**Figure 4 Validation and evaluation of GRS-TME classifier in different cohorts and clinical subgroups.** (A) Kaplan–Meier survival analysis of the GRS-TME classifier in the CGGA cohort; (B) ROC analysis of the GRS-TME classifier in the CGGA cohort; (C and D) Kaplan–Meier survival analysis of the GRS-TME classifier in the CGGA-325 and CGGA-693 cohorts, respectively; (E and F) Kaplan–Meier survival analysis of the GRS-TME classifier in IDH wild-type and mutant type, respectively; (G and I) Kaplan–Meier survival analysis of the GRS-TME classifier in different WHO grades, respectively.

serves as a crucial prognostic factor for patients with glioma, and the GRS-TME classifier exhibited significant prognostic differences across various WHO grades ($p < 0.05$; Figs. 4G and 4I). Univariate and multivariate Cox analysis demonstrated that GRS-TME classifier was also independent prognostic factors for overall survival (Table 1). These findings

**Table 1 Univariate and multivariate Cox regression analysis.**

| Variables | Univariate analysis | | Multivariate analysis | |
|---|---|---|---|---|
| | HR [95% CI] | *p*-value | HR [95% CI] | *p*-value |
| Age | 1.029 [1.021–1.036] | <0.001 | 1.012 [1.004–1.019] | 0.002 |
| Gender | | | | |
| Female | 1.000 | | NA | NA |
| Male | 1.022 [0.869–1.202] | 0.790 | NA | NA |
| Grade | | | | |
| II | 1.000 | | 1.000 | |
| III | 2.784 [2.158–3.592] | <0.001 | 2.257 [1.673–3.046] | <0.001 |
| IV | 7.682 [6.001–9.834] | <0.001 | 4.327 [3.16–5.924] | <0.001 |
| IDH mutation | | | | |
| Wildtype | 1.000 | | 1.000 | |
| Mutant | 0.331 [0.28–0.391] | <0.001 | 0.904 [0.714–1.146] | 0.405 |
| 1p19q codeletion | | | | |
| Non-codeletion | 1.000 | | 1.000 | |
| Codeletion | 0.230 [0.175–0.302] | <0.001 | 0.363 [0.263–0.500] | <0.001 |
| MGMTp methylation | | | | |
| Un-methylated | 1.000 | | 1.000 | |
| Methylated | 0.810 [0.683–0.961] | 0.016 | 0.864 [0.717–1.041] | 0.125 |
| Recurrence status | | | | |
| Non-recurrence | 1.000 | | 1.000 | |
| Recurrence | 2.005 [1.697–2.367] | <0.001 | 2.082 [1.713–2.529] | <0.001 |
| GRS-TME | | | | |
| GRS$^{Low}$/TME$^{High}$ | 1.000 | | 1.000 | |
| Mixed | 2.082 [1.676–2.587] | <0.001 | 1.546 [1.181–2.023] | 0.001 |
| GRS$^{High}$/TME$^{Low}$ | 3.965 [3.204–4.906] | <0.001 | 1.812 [1.378–2.382] | <0.001 |

**Note:**
NA, Not applicable.

further demonstrated the prognostic significance of the GRS-TME classifier in various cohorts and clinical subgroups for glioma patients.

## Molecular characteristics of the GRS-TME classifier

Subsequently, we analyzed gene mutations to enhance our comprehension of the molecular characteristics of the distinct GRS-TME subgroups in glioma. The top 15 genes with the highest mutation rates were identified in the GRS$^{low}$/TME$^{high}$ and GRS$^{high}$/TME$^{low}$ subgroups, with *IDH1*, *TP53*, and *ATRX* having mutation rates exceeding 10%. In the GRS$^{low}$/TME$^{high}$ subgroup, four genes with the highest mutation frequencies were *IDH1* (93%), *TP53* (50%), *ATRX* (45%), and *CIC* (25%), whereas in the GRS$^{high}$/TME$^{low}$ subgroup, they were *TP53* (34%), *TTN* (25%), *EGFR* (25%), and *PTEN* (25%) (Figs. 5A and B). Moreover, we investigated the association between tumor mutational burden (TMB) and the GRS-TME classifier. The results revealed that the GRS$^{low}$/TME$^{high}$ subgroup had the lowest TMB, whereas the GRS$^{high}$/TME$^{low}$ subgroup exhibited the highest TMB

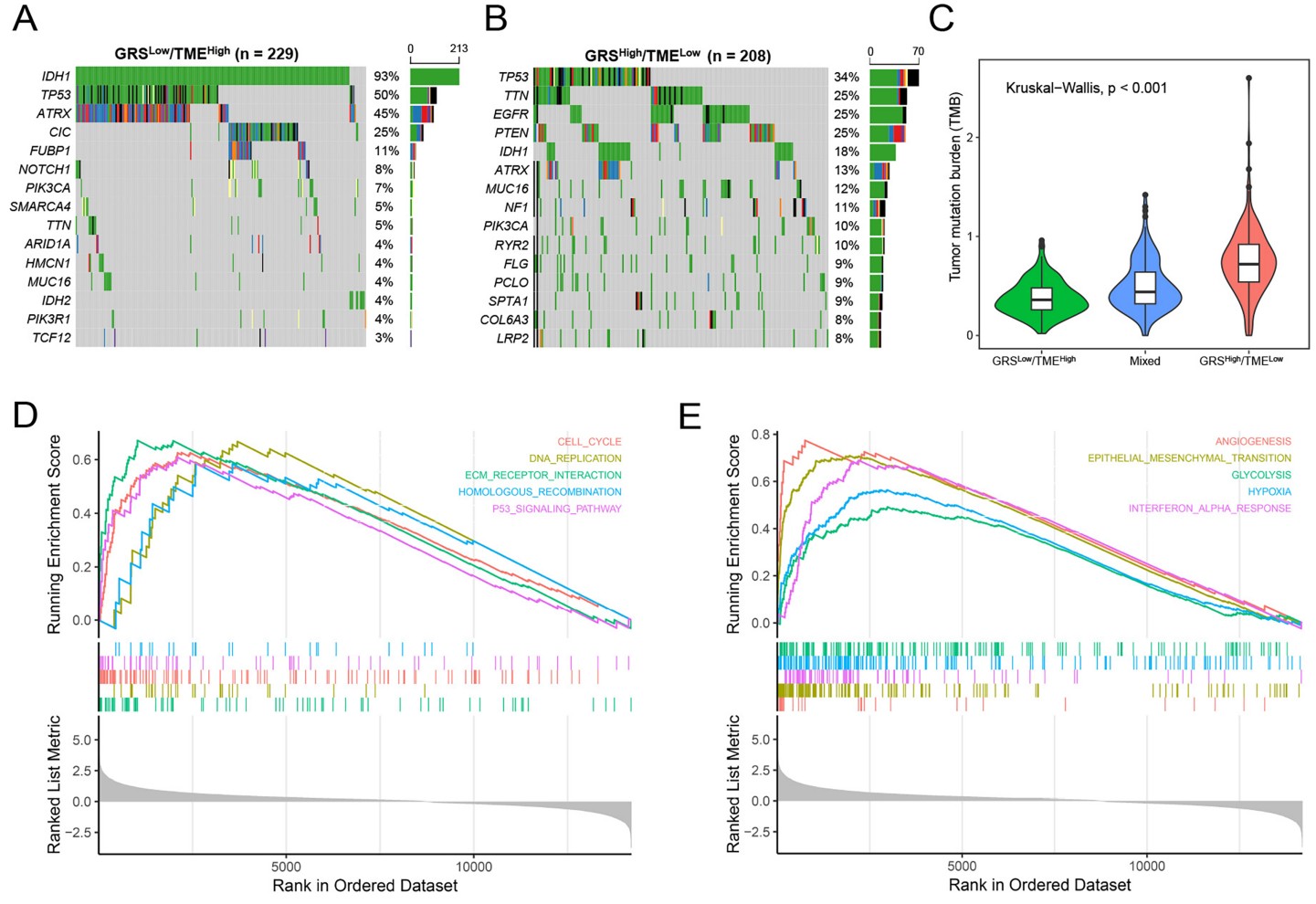

**Figure 5 Molecular features of the GRS-TME classifier in glioma.** (A and B) The top 15 mutation genes in the the GRS$^{low}$/TME$^{high}$ and GRS$^{high}$/TME$^{low}$ subgroups, respectively; (C) comparison of tumor mutational burden among different GRS-TME classifier subgroups; (D and E) KEGG and cancer-related hallmarks signaling pathways enrichment analysis, respectively.

($p < 0.001$; Fig. 5C). Moreover, we employed GSEA to examine the intratumor cellular signaling pathways within the GRS-TME classifier. The results of KEGG enrichment analysis demonstrated significant enrichment of the cell cycle, DNA replication, ECM receptor interaction, homologous recombination, and P53 signaling pathway in the GRS$^{high}$/TME$^{low}$ subgroup (Fig. 5D). Additionally, the enrichment analysis of cancer-related hallmarks indicated significant enrichment of angiogenesis, epithelial-mesenchymal transition, glycolysis, hypoxia, and interferon-alpha response in the GRS$^{high}$/TME$^{low}$ subgroup (Fig. 5E).

## Prediction of immunotherapy response based on the GRS-TME classifier in glioma

To assess the predictive value of the GRS-TME classifier in immunotherapy response, we initially analyzed the expression levels of activation and inhibitory immune markers in distinct subgroups of the GRS-TME classifier. Clear evidence indicates the presence of

distinct expression patterns of immune molecules among various subgroups, encompassing activation immune markers like *CD28, CD40, CXCR4*, and *IL6* (Fig. 6A). In the GRS$^{high}$/TME$^{low}$ subgroup, the inhibitory immune checkpoint molecules (*CTLA4, IDO1, LAG3,* and *PDCD1*) exhibited elevated expression levels compared to the GRS$^{low}$/TME$^{high}$ subgroup (Fig. 6B). Furthermore, we employed the TIDE algorithm to predict T cell dysfunction and exclusion, as well as the response rates to immunotherapy in glioma. We found substantial differences in TIDE and T-cell exclusion score across various GRS-TME subgroups, with the highest values observed in the GRS$^{high}$/TME$^{low}$ subgroup (Figs. 6C and 6D). Furthermore, the immunotherapy responsive group displayed lower GRS score and higher TME score compared to the non-responsive group to immunotherapy (Figs. 6E and 6F). Lastly, we validated the predictive value of the GRS-TME classifier in immunotherapy response using the IMvigor210 immunotherapy cohort. In line with the previous findings, the GRS$^{low}$/TME$^{high}$ subgroup demonstrated the highest immunotherapy response rate and the most favorable survival prognosis ($p < 0.05$; Figs. 6G and 6H). These results indicate the capacity of the GRS-TME classifier to predict the immunotherapy response in patients with glioma.

### KIF20A as a prognostic biomarker and therapeutic target for glioma

KIF20A was identified as a key gene in glioma by using protein-protein interaction (PPI) network. Previous research has indicated that elevated expression of KIF20A is associated with an unfavorable prognosis in patients with glioma. Nevertheless, the role and mechanism of KIF20A in glioma remain unclear. We firstly examined the expression of KIF20A in glioma, and we observed a significant upregulation of KIF20A in tumor tissue compared to normal tissue in the TCGA-LGG and TCGA-GBM cohorts (all $p < 0.05$, Fig. 7A). Furthermore, we discovered that glioma patients exhibiting higher expression levels of KIF20A had a worse prognosis than those with lower expression levels (all $p < 0.001$, Figs. 7B and 7C). As shown in Figs. 7D and 7E, siKIF20A significantly decreased the expression of KIF20A in U251 cells compared to the siNC group. Importantly, our *in vitro* experimental results demonstrated that siRNA targeting KIF20A effectively suppressed the proliferation and migration of U251 cells (Figs. 7F and 7G). Collectively, our study suggested that KIF20A is a potential biomarker for glioma.

## DISCUSSION

Glioma is a complex and aggressive malignancy characterized by a challenging prognosis, which presents difficulties in establishing effective clinical treatment strategies. Consequently, additional research is necessary to advance our comprehension of the underlying molecular mechanisms in glioma and to devise enhanced prognostic assessment and treatment strategies. In this study, we conducted transcriptome analysis of four glioma cohorts and identified glycolysis as the foremost prognostic risk factor in glioma. Analysis of the immune microenvironment in glioma revealed M2 macrophages as the predominant immune cells, while also demonstrating that glycolysis promotes the formation of an immunosuppressive microenvironment and immune evasion in glioma. Considering the significance of glycolysis and the tumor microenvironment in glioma

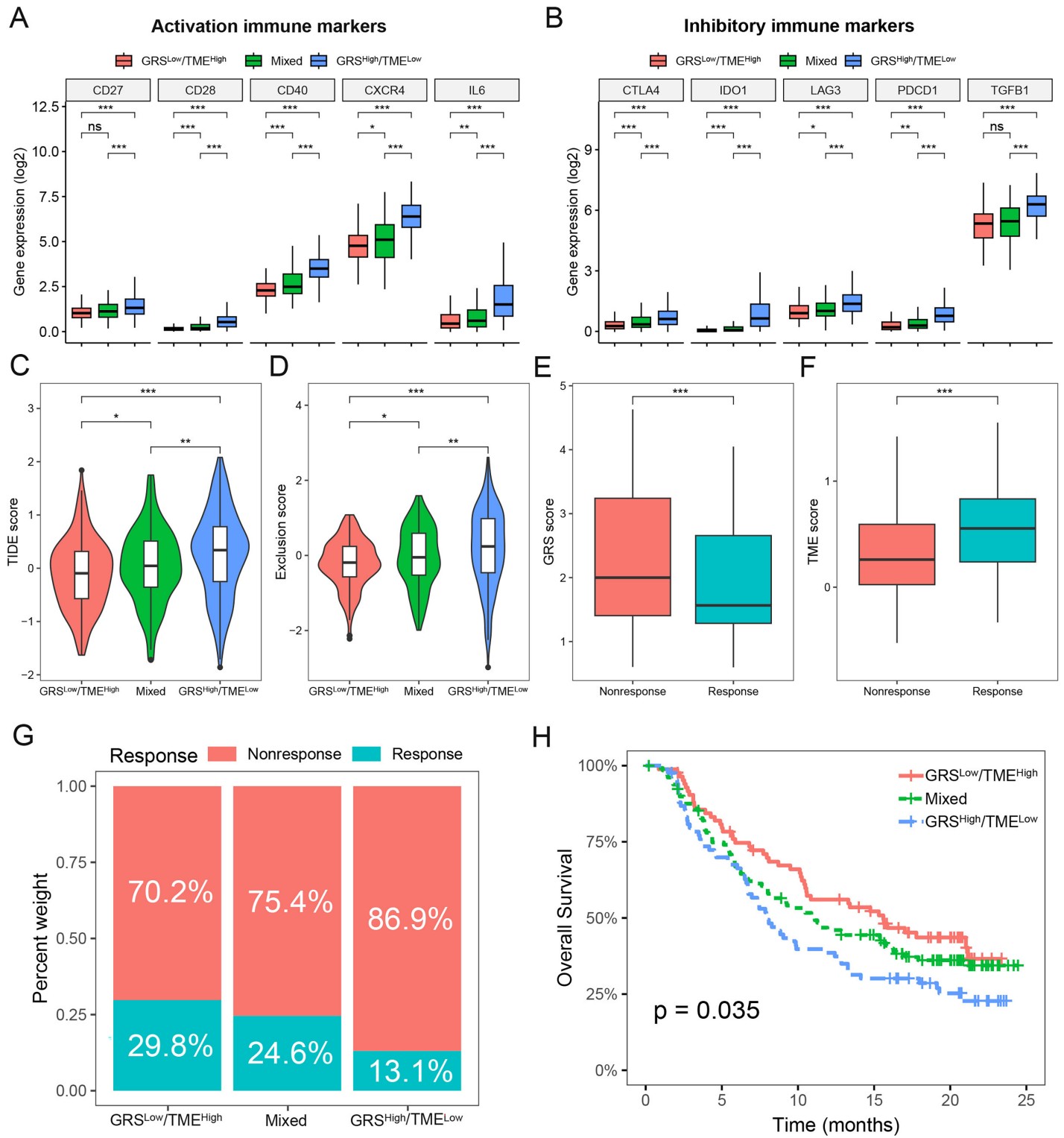

**Figure 6 Prediction of immunotherapy response based on GRS-TME classifier in glioma.** (A and B) The expression of activation and inhibitory immune markers in different GRS-TME subgroups, respectively; (C and D) TIDE, and T-cell exclusion score in different GRS-TME subgroups, respectively; (E and F) GRS and TME score in different GRS-TME subgroups, respectively; (G) the different percentages of anti-PD-L1 immunotherapy in the IMvigor210 cohort; (H) Kaplan–Meier survival analysis of the GRS-TME classifier in the IMvigor210 cohort. $^*p < 0.05$; $^{**}p < 0.01$; $^{***}p < 0.001$; ns, not significant.

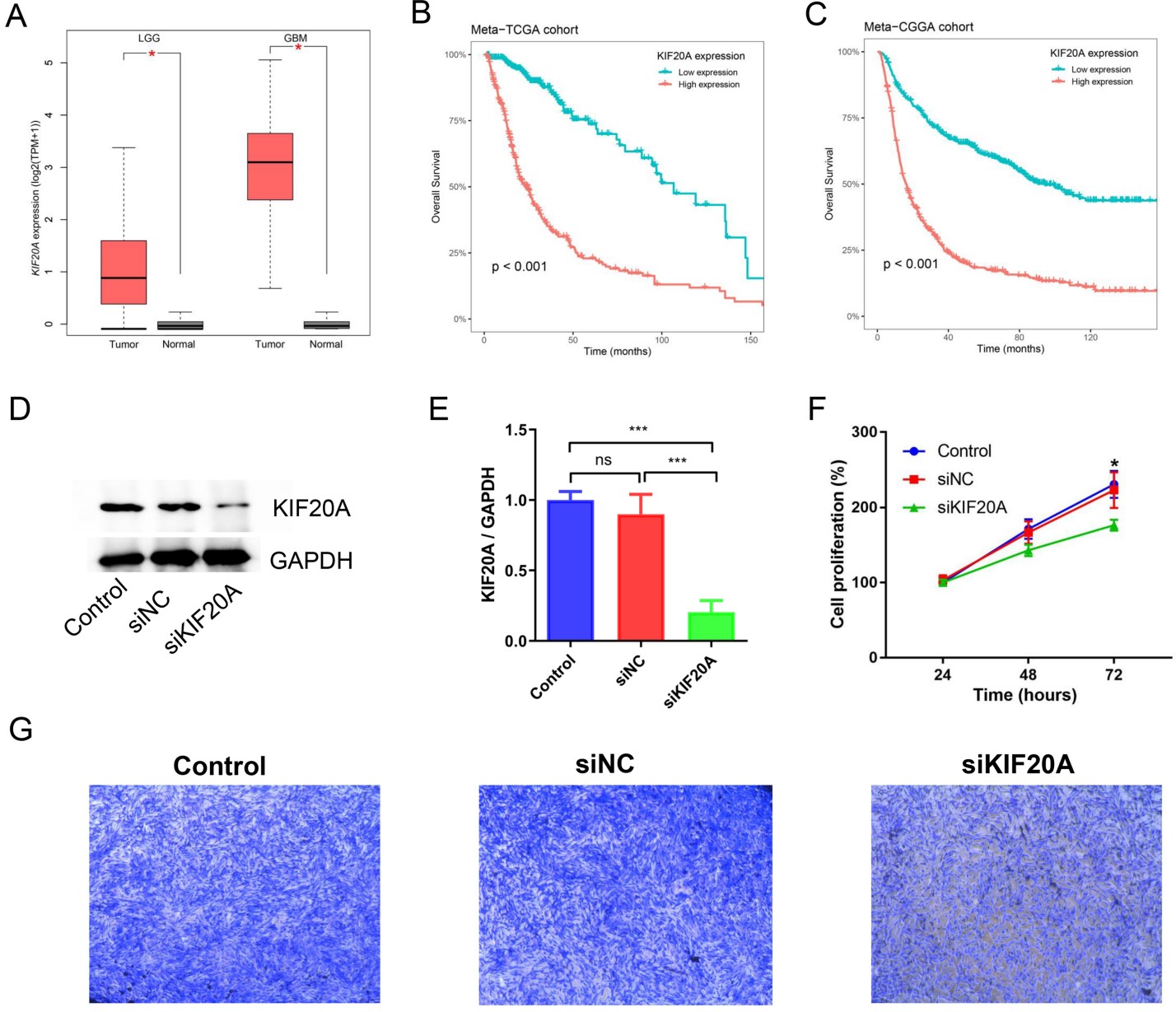

**Figure 7 KIF20A as a prognostic biomarker and therapeutic target for glioma.** (A) The relative expression of KIF20A in TCGA-LGG and TCGA-GBM cohorts; (B) Kaplan–Meier survival analysis of the KIF20A in the Meta-TCGA cohort; (C) Kaplan–Meier survival analysis of the KIF20A in the Meta-CGGA cohort; (D and E) The efficiency of siKIF20A was verified by Western blot; (F) KIF20A knock down can significantly inhibit U251 cell proliferation; (G) KIF20A knock down can significantly inhibit U251 cell migration. $^{*}p < 0.05$; $^{***}p < 0.001$.

prognosis, we developed a novel classifier named GRS-TME for the assessment of glioma prognosis. We validated the GRS-TME classifier using various cohorts and clinical subgroups (IDH mutant and WHO Grade), demonstrating its predictive value for immunotherapy response in patients with glioma. Furthermore, this study identified the significant role of the KIF20A gene in glioma through *in vitro* experiments. KIF20A is a kinesin motor protein with significant roles in cell division and intracellular transport. It is

overexpressed in various cancers, including glioma, breast cancer, lung cancer, and colorectal cancer. The overexpression of KIF20A can contribute to cancer cell proliferation, migration, and invasion. Elevated KIF20A levels have been associated with advanced tumor grades and poorer prognosis in patients with glioma. These findings highlight its potential as a prognostic marker and therapeutic target in glioma.

Glycolysis is a crucial energy metabolism pathway that converts glucose into lactate, generating energy under hypoxic conditions (*Abdel-Wahab, Mahmoud & Al-Harizy, 2019*). Enhanced glycolysis provides sufficient energy and metabolic products for glioma cells, promoting tumor cell growth, invasion, and metastasis (*Paul, Ghosh & Kumar, 2022*). Glioma universally exhibit increased activity of the glycolytic pathway, which is closely linked to unfavorable prognosis and treatment resistance. Furthermore, glycolysis significantly influences the regulation of the immune microenvironment (*Arner & Rathmell, 2023*). Increased glycolysis induces the generation of substantial lactate by glioma cells, thereby influencing the function and activity of immune cells. The accumulation of lactate inhibits immune cell function, suppresses T cell proliferation and activation, reduces natural killer cell cytotoxicity, and promotes the increase of immune suppressive cells, including regulatory T cells and macrophages (*Ye, Jiang & Zhang, 2022*). Hence, glycolysis has the potential to serve as a biomarker for prognostic assessment in patients with glioma. Through a meta-analysis of four glioma transcriptome cohorts, we determined that glycolysis is the most important prognostic risk factor for glioma patients. Glioma patients with lower levels of glycolysis have a more favorable prognosis in the TCGA and CGGA cohorts compared to those with elevated levels of glycolysis. These findings suggest that targeting glycolysis has become a critical therapeutic strategy for patients with glioma.

Glioma is a heterogeneous tumor characterized by diverse cell types and molecular subtypes, which can influence the development of the tumor immune microenvironment (*DeCordova et al., 2020*). Subsequently, we performed a comprehensive analysis of the immune microenvironment in glioma patients. The results indicated a correlation between glioma heterogeneity and glycolysis, as gliomas with high glycolysis levels exhibited lower tumor purity compared to those with low levels. In addition, we found a significant association between glycolysis and an immunosuppressive microenvironment characterized by increased levels of immune suppressor cells, such as CAFs and TAMs. CAFs and TAMs play important roles in the growth, invasion, metastasis, and immune response of glioma by participating in multiple signaling pathways (*Gunaydin, 2021*). Activated CAFs can generate abundant extracellular matrix ECM components, including collagen protein, thus enhancing the adhesion and migratory capacities of tumor cells. CAFs can also participate in tumor cell proliferation, invasion, and metastasis through other signaling pathways such as Wnt, Hedgehog, and Notch (*Fang et al., 2023*). Meanwhile, activated TAMs can secrete various cytokines and chemicals, such as IL-10, TGF-β, and VEGF, which inhibit the function of immune cells, regulate immune responses, and promote tumor growth (*Mao et al., 2021*). Furthermore, we found that

lower levels of activated NK cells and helper T cells in glioma patients with high glycolysis levels. Helper T cells can stimulate the activation of immune cells *via* IFN-γ, thereby activating macrophages and the cytotoxicity of NK cells, and facilitating antigen presentation by antigen-presenting cells like dendritic cells. Moreover, helper T cells can secrete cytokines such as interleukin-2 and interleukin-17, which regulate the activation of immune cells. These findings suggest that glycolysis can contribute to immune evasion in glioma, thereby enhancing our understanding of the involvement of glycolysis in the immune microenvironment of glioma.

Currently, specific gene expression signatures can be used to predict the survival prognosis of glioma patients. Due to the significant heterogeneity of glioma, it is imperative to identify novel biomarkers that can improve the precision of clinical prognostic predictions only based on prognostic-related genes. Considering the importance of glycolysis and the TME as prognostic factors in glioma, we developed a GRS-TME classifier by combining the glycolysis (18 genes) and TME (seven immune cells) scores in this study. Patients with low GRS score or high TME score had better overall survival outcomes. Glioma patients were classified into four subgroups based on their GRS score and TME score: GRS$^{low}$/TME$^{high}$, GRS$^{low}$/TME$^{low}$, GRS$^{high}$/TME$^{high}$, and GRS$^{high}$/TME$^{low}$. Survival analysis of the GRS-TME classifier revealed significant differences across multiple independent cohorts, with the GRS$^{low}$/TME$^{high}$ subgroup demonstrating the best survival prognosis compared to other subgroups.

We further evaluated the role of the GRS-TME classifier in immunotherapy. Using the TIDE algorithm, we found that GRS$^{high}$/TME$^{low}$ subgroup patients had higher T-cell dysfunction and exclusion score, while GRS$^{low}$/TME$^{high}$ subgroup patients were more likely to have a higher response rate to immunotherapy. Moreover, we found that significant differences in the expression levels of immune checkpoints among the GRS-TME subgroups, such as IDO1, LAG3, and PDCD1, suggesting that different subgroups might exhibit distinct response rates to immunotherapy. We performed a comprehensive investigation of the role of GRS-TME classifier in the IMvigor210 cohort, and found that patients in the GRS$^{low}$/TME$^{high}$ subgroup had a higher response rate to immunotherapy and clinical survival outcomes. These results suggest that the GRS-TME classifier has the ability to identify the responsive population for immunotherapy, and has the potential to serve as a novel biomarker for immunotherapy in patients with glioma.

However, there are several limitations in this study. First, although we conducted validation in multiple cohorts, further validation of these results is still required in independent cohorts. Second, this study relied on bioinformatics methods and publicly available databases for analysis, while the actual situations in clinical research may be influenced by other factors. Therefore, future clinical studies should consider more patient characteristics and clinical variables. In addition, despite the preliminary functional investigation of the KIF20A in this study, its precise mechanism of action in glioma remains unclear, demanding further comprehensive research to clarify its molecular mechanism.

## CONCLUSIONS

This study developed and validated a personalized classifier based on glycolysis and tumor microenvironment to aid in the prediction of the survival prognosis of glioma patients, which may help to guide clinical decisions.

## ACKNOWLEDGEMENTS

We thank Dr. Jianming Zeng (University of Macau), and all the members of his bioinformatics team, biotrainee, for generously sharing their experience and codes. All authors are also grateful to patients who contributed data to this study.

### Funding

The authors received no funding for this work.

### Competing Interests

The authors declare that they have no competing interests.

### Author Contributions

- Pengfei Fan conceived and designed the experiments, performed the experiments, prepared figures and/or tables, and approved the final draft.
- Jinjin Xia performed the experiments, prepared figures and/or tables, and approved the final draft.
- Meifeng Zhou analyzed the data, authored or reviewed drafts of the article, and approved the final draft.
- Chao Zhuo analyzed the data, authored or reviewed drafts of the article, and approved the final draft.
- Hui He conceived and designed the experiments, prepared figures and/or tables, and approved the final draft.

### Data Availability

The raw data and code is available in the Supplemental File.

### Supplemental Information

Supplemental information for this article can be found online at http://dx.doi.org/10.7717/peerj.16066#supplemental-information.

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
