# Peer review of "Development and validation of a personalized classifier to predict the prognosis and response to immunotherapy in glioma based on glycolysis and the tumor microenvironment"

_PeerJ, doi:10.7717/peerj.16066_

## Round 0.1 · original submission · Minor Revisions

Please revise the manuscript as suggested.

Reviewer 1 ·

Basic reporting

no comment

Experimental design

no comment

Validity of the findings

no comment

Additional comments

1) First, there are some prognostic models for glioma patients have been developed. In comparison with previous studies, what is the primary distinguishing feature of this study? This should be discussed.
2) Second, authors should supplement the results section with the results of univariate and multivariate Cox regression analysis. This is beneficial for assessing whether the GRS-TME classifier was independent prognostic factors.
3) Third, the introduction of the main text is not adequate, please gave a brief review on known biomarkers associated with the prognosis of glioma and treatment response of the immunotherapy, and indicate the potential strengths of GRS-TME classifier, to support the needs for this research.
4) Please report the threshold value of C-index for a good prediction model in both the training and validation samples. In statistics, please ensure P<0.05 is two-sided.
5)Fifth, the discussion of the main text is not adequate, please gave a brief review on KIF20A, and other key genes.

Reviewer 2 ·

Basic reporting

The paper titled “Development and validation of a personalized classifier to predict the prognosis and response to immunotherapy in glioma based on glycolysis and the tumor microenvironment” is interesting. The GRS-TME classifier might be a beneficial tool to aid in the prognostic evaluation and risk stratification of osteosarcoma patients. However, there are several minor issues that if addressed would significantly improve the manuscript.
1) What are the biggest advantages and disadvantages of the GRS-TME classifier in this study? It is recommended to add relevant contents.
2) The introduction part of this paper is not comprehensive enough, and the similar papers have not been cited. It is recommended to quote relative article.
3) There have been many studies on glioma. What is the difference between this study and previous studies? These need to be described in the introduction.
4) In addition to the GRS-TME classifier in this study, what better tools can be used for prognosis assessment and risk stratification of glioma patients? It is recommended to add relevant contents.

Experimental design

In this study, bioinformatics approaches were employed to develop the prognostic model. It is suggested to add further functional experiments to study its role in vivo and potential molecular mechanisms.

Validity of the findings

no comment

Additional comments

no comment

Reviewer 3 ·

Basic reporting

Metabolism and tumor microenvironment are important determinants of prognosis and immunotherapeutic efficacy for cancer patients. In the manuscript “Development and validation of a personalized classifier to predict the prognosis and response to immunotherapy in glioma based on glycolysis and the tumor microenvironment”, authors developed a personalized classifier based on glycolysis and the tumor microenvironment to effectively predict prognosis and immunotherapeutic response in patients with glioma. There are several minor issues need to be addressed.

(1) There were several similar reports (Front Genet. 2022 Jun 2;13:899125.) and Front Genet. 2020 Apr 15;11:363.) about the identification of an immune signature in glioma in PubMed. What is the novel idea in the paper? Please elaborate in the introduction.

(2) Some studies have already reported prognostic models in the glioma. Please add relevant descriptions in the introduction and explain the differences between this study and other research.

(3) What are the associations between glycolysis and immunotherapeutic response?

(4) This study showed that siRNA targeting KIF20A effectively suppressed the proliferation and migration of glioma cells. Authors should supplement a brief discussion about the KIF20A in the glioma.

(5) Although the GRS-TME classifier was validated and evaluated in different cohorts and clinical subgroups. It is suggested that the results of the multivariate Cox regression analysis of the GRS-TME classifier should be analyzed.

(6) This study focuses on development and validation of a personalized classifier based on glycolysis and the tumor microenvironment. However, this study lacks a description of glycolysis-related genes and tumor microenvironment-related cells.

Experimental design

no comment

Validity of the findings

no comment

Additional comments

no comment

---

## Round 0.2 · accepted · Accept

This manuscript can be accepted now.